# Analysis of the deployment scale and investment prediction of China's coal power carbon capture technology under typical scenarios before 2050

**Jianfang Zong** [1]*, **Qing Ding²**, **Haitao Wang¹**, **Xinyu Cao¹**, **Jie Wei¹**

1 Division of General Standardization, China National Institute of Standardization, Beijing, China,
2 Department of Resource and Environment Branch, China National Institute of Standardization, Beijing, China

¤ Current address: China National Institute of Standardization, Beijing, China
* 15318866884@qq.com

## Abstract

This study evaluates the potential for green and low-carbon transformation in China's coal-fired power sector by analyzing seven representative scenarios, including projections for total installed capacity, power generation, and coal-fired power metrics before 2050. Carbon emissions are estimated using a conversion factor of 0.899 kg $CO_2$ per kWh, while deployment scales and emission reduction potentials for carbon capture technologies are calculated based on IEA (2022) assumptions for low (30%) and high (90%) installation ratios. The study further quantifies installation and operational costs, investment requirements by 2050, and annual investments from 2030 to 2050 under both scenarios. Results demonstrate that high-proportion carbon capture deployment can significantly accelerate the sector's low-carbon transition, fostering an emerging industry worth over 20 trillion yuan and delivering dual benefits of emissions reduction and economic growth. Policy support for large-scale implementation is critical. By pioneering multi-scenario modelling, this research systematically projects China's coal-fired CCUS deployment needs under 1.5°C and 2°C climate targets, integrating IEA frameworks with domestic policy scenarios to assess industrial-scale potential and inform decision-making. This work provides the first systematic integration of IEA projections with China-specific policy scenarios, offering policymakers a framework to balance climate targets with energy transition realities.

## 1. Introduction

In September 2020, during the 75th United Nations General Assembly, China announced its "dual carbon" goals: achieving carbon peaking by 2030 and carbon neutrality by 2060 [1]. As of 2020, the electricity sector accounted for 35.71% of China's carbon dioxide emissions—the highest share among all industries [2]. Carbon capture, utilization, and storage (CCUS) technology is critical for enabling low-carbon

**Data availability statement:** All relevant data are within the paper and its Supporting information files.

**Funding:** The author(s) received no specific funding for this work.

**Competing interests:** The authors have declared that no competing interest exist.

fossil fuel use and reducing emissions in coal-fired power and other industrial processes [3]. According to the Intergovernmental Panel on Climate Change (IPCC), most climate mitigation pathways cannot limit global warming to 1.5°C or 2°C without large-scale CCUS deployment [4]. The International Energy Agency (IEA) further highlights that CCUS could contribute 32% of the emissions reductions needed to meet the 1.5°C target by 2100 [5].Global CCUS adoption is projected to expand dramatically, with annual $CO_2$ capture increasing from about 40 million tons in 2020 to about 7.6 billion tons by 2050—a 100-fold rise [6]. Under IEA's Sustainable Development Scenario, CCUS will provide 15% of cumulative emissions cuts required for global net-zero by 2070 [6]. The Global Energy Internet Development Cooperation Organization has identified CCUS as a foundational technology for China's carbon peaking and neutrality goals, emphasizing its role in energy and power planning through 2060 [7].

China has prioritized CCUS development through policy and innovation. The Action Plan for Carbon Peak Before 2030 advocates pilot projects in hydrogen metallurgy and integrated $CO_2$ capture/utilization, alongside advancements in low-cost capture and storage [8]. The 14th Five-Year Plan (2021–2025) designates CCUS as a key demonstration project, while 2022 guidelines from the National Development and Reform Commission (NDRC) and National Energy Administration (NEA) call for strengthened R&D support and policy frameworks for CCUS in thermal power [9]. In 2023, the Ministry of Science and Technology's Carbon Peaking/Carbon Neutrality Technology Innovation Action Plan further classified CCUS as a "cutting-edge disruptive technology" [10].

Carbon Capture, Utilization and Storage (CCUS) encompasses technologies that capture $CO_2$ from industrial processes, fossil fuel combustion, or direct air capture, followed by geological storage or conversion into valuable products. The technology chain comprises four key components: (1) capture via pre-combustion, post-combustion, or oxy-fuel methods; (2) transportation through pipelines, ships, or tankers; (3) utilization through geological, chemical, or biological pathways; and (4) permanent geological storage. This integrated approach enables both emission reduction and resource recovery from $CO_2$ streams.

Geological Utilization Pathways: Geological applications demonstrate significant scale potential, with two primary methods. Enhanced Oil Recovery (EOR), as implemented in China's Jilin Oilfield project (300,000 tons $CO_2$/year stored with 15% oil yield increase), and deep saline aquifer storage exemplified by Norway's Northern Lights project (current 1.5 million tons/year capacity, expanding to 5 million). These methods provide immediate emission reductions while creating economic value through either hydrocarbon production or secure long-term sequestration.

Chemical conversion applications: Chemical utilization transforms $CO_2$ into commercial products through catalytic processes. Notable examples include Iceland's geothermal-powered methanol synthesis (4,000 tons/year) and China's Ningxia coal-to-urea project (500,000 tons $CO_2$/year). Advanced materials production is demonstrated by Germany's Covestro, manufacturing polycarbonates with 0.5 tons $CO_2$ fixed per ton product. These applications showcase the potential for carbon circularity in chemical manufacturing sectors.

Biological utilization methods: Biological pathways offer dual environmental benefits through microalgae cultivation (e.g., U.S.-based Algenol's 200 tons $CO_2$/hectare/year biofuel production) and agricultural enhancement (Dutch greenhouse operations achieving 20%~30% yield increases at 800–1000 ppm $CO_2$). These nature-based solutions complement engineered approaches while supporting food and energy production systems.

Storage refers to the permanent sequestration of $CO_2$ in deep geological formations, isolating it from the atmosphere. As the only technology enabling low-carbon fossil fuel use and large-scale emissions reduction, CCUS plays a critical role in climate mitigation strategies [7].

Coal-fired CCUS technology captures and purifies $CO_2$ emissions from power plants for subsequent utilization or permanent storage, with post-combustion capture currently dominating engineering applications. While utilization and storage pathways remain diverse with uncertain market adoption, this study focuses on system-level assessment of three critical dimensions: (1) capture capacity, (2) emission reduction potential, and (3) investment requirements. Projections indicate China's coal-power CCUS could achieve 2.9 billion tons annual $CO_2$ reduction by 2050 - representing over 40% of global capacity (IEA's 3 billion ton estimate) - with retrofit costs ($1,000~3,000/kW) being 15%~20% lower than Western benchmarks. However, economic viability requires substantial policy support, particularly carbon pricing exceeding 200 yuan/ton, alongside solutions for grid integration challenges arising from capture systems' 20%~30% energy penalty.

On cost structures and research gaps, existing studies analyze coal-CCUS through three lenses: technological development, economic feasibility, and environmental impacts. The comprehensive cost structure comprises [8]: (1) economic costs (capital expenditures: $1,000~3,000/kW installation; operational expenses: capture, transport, storage); and (2) environmental costs (15%~30% energy penalty; 0.1%~0.5% annual leakage risk) [9]. Current research highlights critical challenges: capture phase dominates system costs (60%~75%), efficiency losses may offset 15%~25% of emission reductions, and storage site availability limits regional deployment. These factors underscore the need for integrated assessments balancing technological potential with real-world constraints in China's energy transition context.

This study employs seven distinct scenarios to analyze China's coal-fired power CCUS development pathways [10–13]: (1) Policy Scenario: Maintains current low-carbon transformation trends aligned with China's Nationally Determined Contributions (NDCs) under the Paris Agreement. (2) Strengthened Policy Scenario: Builds upon Policy Scenario with Strengthened GDP energy intensity reductions (30%~35% below 2005 levels by 2030), and increased non-fossil energy share (>25% primary energy by 2030), with tighter $CO_2$ emission controls. (3) 2°C Scenario: Models pathways consistent with limiting global warming to 2°C, including deep decarbonization strategies, technology deployment roadmaps, and policy and financial requirements. (4) 1.5°C Scenario: Analyzes net-zero $CO_2$ pathways by mid-century, assessing: Technical feasibility, Socioeconomic impacts, Greenhouse gas reduction potentials. (5) Literature-Based Scenario [12]: Incorporates projections from China's Energy and Electricity Development Plan (2030–2060). (6) Stated Policies Scenario (SPS): Reflects current policy trajectories based on Global energy market trends, Technology cost projections, Existing national policy frameworks. (7) Announced Pledges Scenario (APS): Assumes full implementation of all national on Climate commitments, Net-zero targets, Energy transition plans

While prior studies address techno-economic aspects of CCUS, systematic projections of deployment pathways, investments, and emission reductions under diverse climate targets remain scarce. This study fills this gap by quantifying CCUS capacity, costs, and mitigation potential across the seven scenarios. Using pre-2050 power system data (installed capacity, generation, and coal-fired share), we project coal-power emissions via a 0.899 kg/kWh emission factor (Shanghai Standard DB31/T 1139-2019). We further estimate 2050 installation/operation costs and annual investments (2030–2050) for low- and high-penetration CCUS, revealing cumulative costs under both scenarios. Results demonstrate that high-penetration CCUS can drive coal-power decarbonization, with investments potentially exceeding 20 trillion yuan, fostering a new green industry while balancing low-carbon transition and economic growth.

This paper establishes three key assumptions. First, a technical assumption specifies that coal-fired CCUS primarily employs post-combustion capture, with costs and efficiency following current trends (e.g., an emission factor of 0.899 kg/

kWh). Second, a policy assumption posits that seven scenarios—such as policy reinforcement and temperature control targets—can encompass China's potential emission reduction pathways. Third, an economic assumption projects that carbon capture investment and operating costs can be extrapolated to 2050 based on existing data. Unlike prior studies focusing on singular technological or economic analyses, this paper employs multi-scenario modeling to systematically forecast the large-scale deployment needs of coal-fired CCUS in China under 1.5°C/2°C targets. By integrating International Energy Agency (IEA) projections with Chinese policy scenarios, this study reveals the potential scale of an emerging CCUS-driven industry, offering novel insights for policymaking.

The deployment of CCUS in China's coal-fired power plants will significantly reshape energy markets and economic dynamics. In the energy sector, coal-fired plants will transition into "flexible peak-shaving power sources," extending their operational lifespan by 10~15 years. However, this will raise power generation costs by 30%~50%, necessitating reforms in electricity pricing mechanisms and catalyzing a trillion-yuan industrial chain encompassing capture equipment and storage services. Economically, while short-term operating costs for coal plants will increase, long-term benefits include new growth opportunities, over one million direct jobs, and the transformation of coal-dependent regions. Additionally, it will enhance China's export competitiveness in low-carbon technologies, particularly for developing countries. Critical challenges include ensuring carbon prices exceed 200 yuan/ton for economic viability, investing in pipeline infrastructure, and dynamically coordinating with renewable energy development.

## 2. Prediction and analysis of the development capacity of carbon capture projects in China's thermal power industry

### 2.1. Development scale of carbon capture in thermal power in China under different scenarios

This study analyzes China's power system in 2050 across seven scenarios: Policy Scenario, Strengthened Policy Scenario, 2°C Scenario, and 1.5°C Scenario (from China's long-term low-carbon strategy reports [10,11]), Literature-Based Scenario [12], and the IEA's Stated Policies Scenario (SPS) and Announced Pledges Scenario (APS) [13,14]. These scenarios integrate domestic and international research to project total installed capacity, coal/gas-fired capacity (GW), and power generation (trillion kWh) for 2050 (Figs 1 and 2). The SPS and APS further incorporate China's total electricity output, coal-fired generation, and gas-fired generation data, providing a comprehensive framework to evaluate energy transition pathways under varying policy and climate targets.

### 2.2. Analysis of carbon capture and emissions in China's coal-fired power industry

To predict $CO_2$ emissions from coal-fired power generation units, this study adopts an advanced benchmarking approach based on the local standard Carbon Emission Indicators for Coal-Fired Power Generation Enterprises (DB31/T 1139-2019). Specifically, we apply the standard's upper-bound emission factor of 899 g $CO_2$/kWh to model sector-wide emissions, ensuring alignment with stringent regional performance benchmarks. This method provides a consistent and replicable framework for assessing emission trajectories under varying operational scenarios.

$$Carbon\ emissions = power\ generation * 0.899\ (kg/kWh)$$
(1)

Table 1 presents the projected 2050 coal-fired power generation and corresponding $CO_2$ emissions for China under seven distinct scenarios: Policy Scenario, Strengthened Policy Scenario, 2°C Scenario, 1.5°C Scenario, Literature-Based Scenario [12], Stated Policies Scenario (SPS), and Announced Pledges Scenario (APS). As technological advancements improve power plant efficiency, we anticipate further reductions in the carbon intensity benchmark. These projections incorporate potential efficiency gains through coefficient adjustments in Formula 1, allowing for dynamic updates to emission estimates as cleaner technologies emerge.

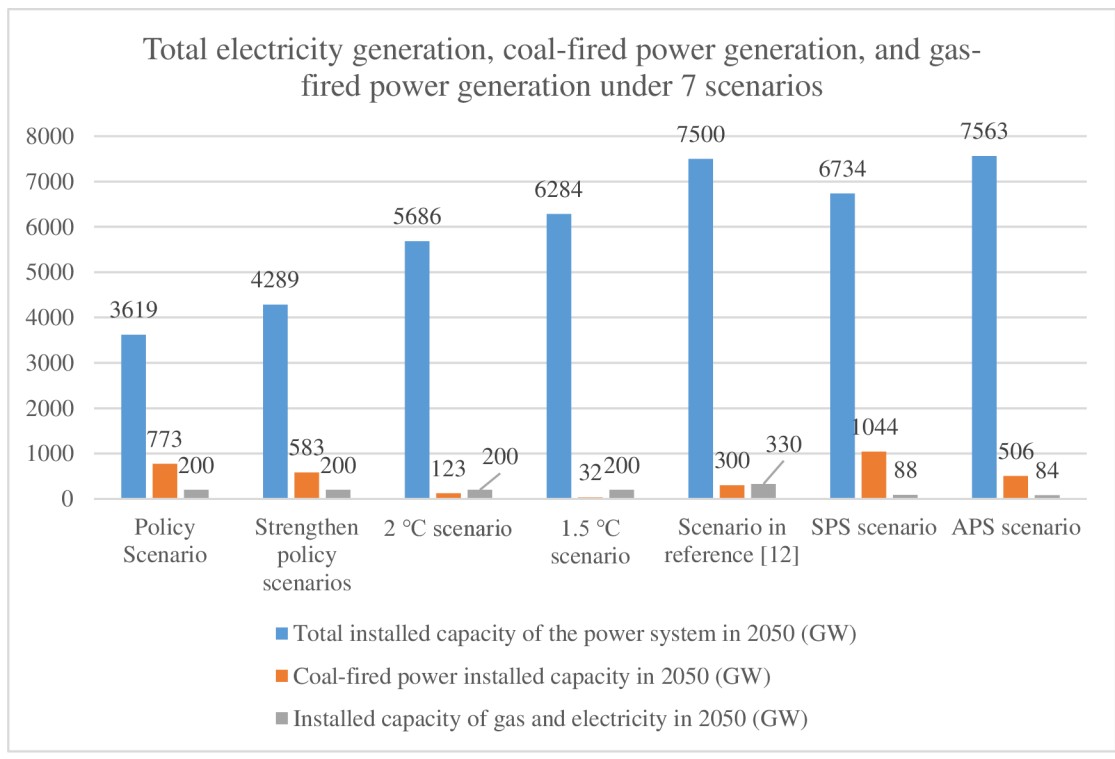

**Fig 1. Projected power generation capacity under seven scenarios in 2050.**

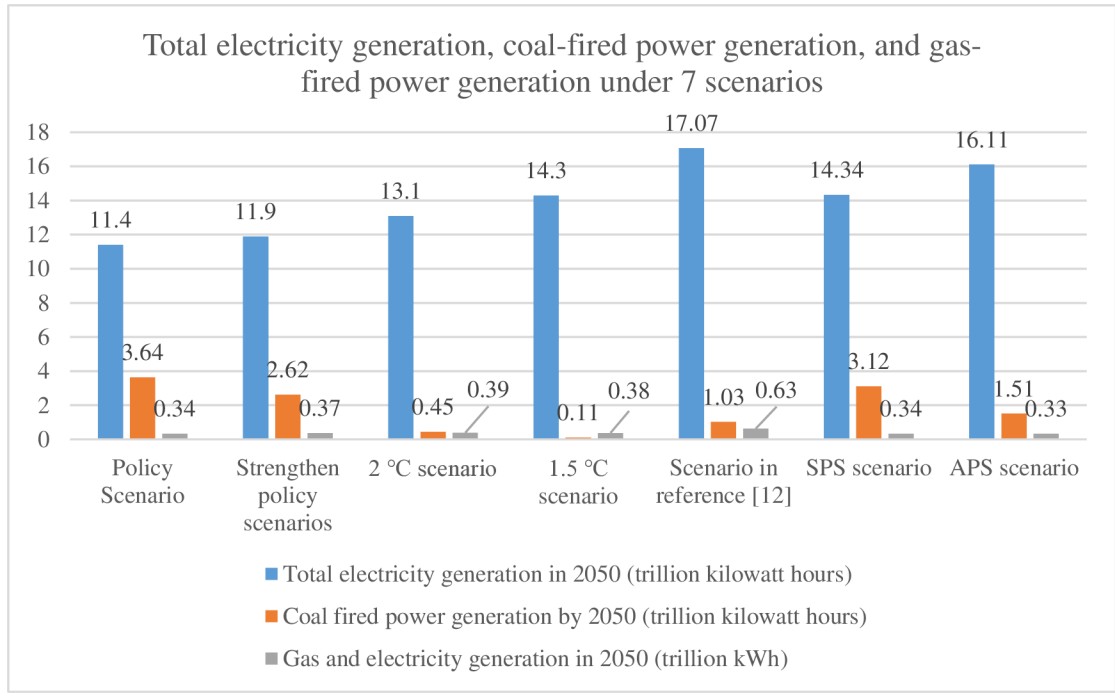

**Fig 2. Electricity generation projections for 2050 across seven scenarios.**

**Table 1. Projected CO$_2$ emissions from coal-fired power generation in China (2050).**

| Scenarios | Policy scenarios | Strengthened policy scenarios | 2°Cscenarios | 1.5°Cscenarios | Literature [12] scenarios | SPS scenario | APS scenario |
|---|---|---|---|---|---|---|---|
| 2050 coal-fired power generation (trillions of kilowatt hours) | 3.64 | 2.62 | 0.45 | 0.11 | 1.03 | 3.12 | 1.51 |
| CO2 emissions from coal-fired power in 2050 (billions of tons) | 32.72 | 23.55 | 4.04 | 0.99 | 9.26 | 28.04 | 13.57 |

## 3. Prediction and analysis of the potential demand for carbon capture and emission reduction in China's coal-fired power industry

China plans to implement CCUS technology extensively across all coal-fired power units between 2050 and 2060 [15]. This delayed large-scale deployment stems from three primary factors: (1) the need for technological maturation, as current demonstration projects remain limited in scale, with megaton-level commercial operations not anticipated before 2045; (2) the optimal retrofit period for existing coal power units, which falls between 2035 and 2045 based on typical renewal cycles; and (3) current policy priorities favoring renewable energy development, which has reduced immediate pressure for CCUS adoption in the coal sector. Additionally, the development of necessary CO$_2$ transport and storage infrastructure requires substantial long-term investment, with significant scale unlikely before 2040. Economically, widespread CCUS implementation will only become viable when carbon prices surpass 200 yuan/ton (projected post-2045) and technology costs decline below 150 yuan/ton. This phased approach mitigates early-stage technological risks while capitalizing on stronger policy incentives as China approaches its 2060 carbon neutrality target.

The IEA report [16] outlines two key decarbonization scenarios for the coal power sector: the Announced Pledges Scenario (APS) and the Net Zero Emissions Scenario (NZE) by 2050. According to these projections, carbon capture adoption rates in coal-fired power plants would reach 30% under APS and 90% under NZE by 2050. Building on these benchmarks, our study establishes two implementation scenarios: a conservative pathway (30% adoption) and an ambitious pathway (90% adoption). For modeling purposes, we assume ideal operating conditions with 100% capture efficiency and no carbon leakage. Table 2 subsequently presents the projected CO$_2$ emission reduction potential for China's coal power industry across seven distinct scenarios in 2050, comparing outcomes under both the 30% (APS-aligned) and 90% (NZE-aligned) carbon capture adoption rates.

$$Low\ adoption\ scenario\ CO2\ emission\ reduction\ potential = CO2\ emissions * 30\% \qquad (2)$$

$$High\ adoption\ scenario\ CO2\ emission\ reduction\ potential = CO2\ emissions * 90\% \qquad (3)$$

According to Reference [9], China's CCUS technology is projected to demonstrate substantial emission reduction potential post-2040, with annual CO$_2$ capture capacity estimated at 370~1,300 million tons by 2040 and 600~1,450 million tons by 2050. The power sector is expected to account for a significant portion of this mitigation potential, contributing approximately 220~600 million tons annually during the 2040~2050 period. These projections highlight the critical role of CCUS in decarbonizing China's coal-dominated power industry while supporting national climate targets.

**Table 2. Carbon capture and CO2 emission reduction potential of China's coal-fired power industry in seven scenarios by 2050 (billion tons).**

| Scenarios | Policy scenarios | Strengthened policy scenarios | 2°Cscenarios | 1.5°Cscenarios | Literature [12] scenarios | SPS scenario | APS scenario |
|---|---|---|---|---|---|---|---|
| Low adoption scenario | 9.82 | 7.07 | 1.21 | 0.30 | 2.78 | 8.41 | 4.07 |
| High adoption scenario | 29.45 | 21.20 | 3.64 | 0.89 | 8.33 | 25.24 | 12.21 |

As shown in Table 2, the high-adoption scenario (90% carbon capture implementation) demonstrates sufficient capacity to achieve China's projected CCUS potential, supporting the nation's carbon neutrality objectives. However, these emission reduction estimates contain inherent uncertainties that primarily depend on the actual deployment rate of carbon capture systems across coal-fired power plants. The realization of this potential will be contingent upon overcoming technical, economic, and policy barriers to large-scale CCUS adoption in the power sector.

While this study assumes ideal 100% capture efficiency for modelling purposes (Table 2), real-world CCUS implementation faces two critical reliability challenges: carbon leakage (typically 5%~15% due to equipment aging and operational variability) and efficiency constraints. To enhance prediction accuracy, we recommend a three-tier verification framework: (1) adoption of EU CCS Directive monitoring standards for annual injection audits, (2) implementation of ISO 27916 carbon sequestration certification protocols, and (3) continuous calibration of models using operational data from demonstration projects (e.g., the Taizhou CCUS facility). Empirical evidence suggests this approach can reduce emission reduction prediction uncertainties to within ±5%, significantly improving the robustness of CCUS deployment projections.

## 4. Prediction and analysis of carbon capture costs in China's coal-fired power industry

### 4.1. Cost prediction and analysis of carbon capture technology installation in China's coal-fired power industry

Carbon capture retrofits in China's coal-fired power sector can be implemented either plant-wide or partially, with installation costs encompassing capture equipment procurement, boiler/turbine upgrades, and pollution control systems [16]. Based on IEA projections of $1000~$3000/kW retrofit costs by 2030 [16], we estimate installation expenses under seven 2050 scenarios, evaluating both low (30%) and high (90%) adoption rates. All costs are calculated in constant 2025 RMB (exchange rate: 1 USD = 6.849 RMB) without inflation adjustment, with full results presented in Table 3. Our methodology allows for future cost reductions through coefficient modifications in Formulas 4–7 while maintaining the core calculation framework. Technological advancements and economies of scale may further decrease these projections, though our analysis conservatively uses current cost benchmarks.

$$
\begin{aligned}
& \textit{Low adoption scenario cost lower limit} \ (100 \ \textit{million yuan}) \\
& = 1000 \ (\textit{US dollars/kW}) \ * \ 6.849 \ (\textit{yuan/USD}) \ * \ 10 \ -8(100 \ \textit{million yuan/yuan}) * \textit{coal} \\
& - \textit{fired power installed capacity} \ * \ 106 \ (\textit{kW/GW}) \ * \ 30
\end{aligned}
\tag{4}
$$

$$
\begin{aligned}
& \textit{Upper cost limit for low adoption scenario} \ (100 \ \textit{million yuan}) \\
& = 3000 \ (\textit{US dollars/kW}) \ * \ 6.849 \ (\textit{yuan/USD}) \ * \ 10 \ -8(100 \ \textit{million yuan/yuan}) * \textit{coal} \\
& - \textit{fired power installed capacity} \ * \ 106 \ (\textit{kW/GW}) \ * \ 30\%
\end{aligned}
\tag{5}
$$

Table 3. Accumulated installation costs of carbon capture for coal-fired power in China under 7 scenarios by 2050.

| Scenarios | Policy scenarios | Strengthened policy scenarios | 2°C scenarios | 1.5°C scenarios | Literature [12] scenarios | SPS scenario | APS scenario |
|---|---|---|---|---|---|---|---|
| 2050 installed capacity of coal-fired power (GW) | 773 | 583 | 123 | 32 | 300 | 1044 | 506 |
| Lower limit of cost for low adoption scenario (in 100 million yuan) | 15883 | 11979 | 2527 | 658 | 6164 | 21451 | 10397 |
| Upper cost limit for low adoption scenario (in 100 million yuan) | 47648 | 35937 | 7582 | 1973 | 18492 | 64353 | 31190 |
| Lower cost limit of high adoption scenario (100 million yuan) | 47648 | 35937 | 7582 | 1973 | 18492 | 64353 | 31190 |
| Upper cost limit for high adoption scenario(100 million yuan) | 142945 | 107810 | 22746 | 5918 | 55477 | 193060 | 93571 |

*High adoption scenario cost lower limit* $(100\ million\ yuan)$
$$= 1000\ (US\ dollars/kW)\ *\ 6.849\ (yuan/USD)\ *\ 10\ -8(100\ million\ yuan/yuan)\ *\ coal$$
$$-\ fired\ power\ installed\ capacity\ *\ 106\ (kW/GW)\ *\ 90\%$$

(6)

*Upper cost limit for high adoption scenario* $(100\ million\ yuan)$
$$= 3000\ (US\ dollars/kW)\ *\ 6.849\ (yuan/USD)\ *\ 10\ -8(100\ million\ yuan/yuan)\ *\ coal$$
$$-\ fired\ power\ installed\ capacity\ *\ 106\ (kW/GW)\ *\ 90\%$$

(7)

## 4.2. Prediction and analysis of carbon capture operating costs in China's coal-fired power industry

The operational costs of carbon capture in coal-fired power plants comprise solvent consumption, chemical reagents, catalysts, waste treatment, and additional staffing requirements for CCUS facility management [16]. Projections indicate that by 2050, the unit operating cost will decline by approximately 70% from 2025 levels to 30~150 yuan/ton $CO_2$ captured [7], with technology-specific variations: pre-combustion capture (30~50 yuan/ton), post-combustion capture (80~150 yuan/ton), and oxy-fuel combustion (90~150 yuan/ton). Given that post-combustion technology currently dominates China's demonstration projects (>90% adoption) due to its maturity and broad applicability [17–19], our study assumes a 2050 technology mix of 90% post-combustion, 7% oxy-fuel, and 3% pre-combustion capture systems. These projections account for anticipated technological advancements while reflecting current implementation trends in the coal power sector.

The projected operating costs of CCUS in China's coal-fired power plants (30~150 yuan/ton $CO_2$ by 2050) exhibit significant variability due to four key factors: (1) Technical route—post-combustion capture (80~150 yuan/ton) incurs the highest costs due to energy-intensive amine solvent regeneration, while pre-combustion capture (30~50 yuan/ton) benefits from coal gasification synergies, and oxy-fuel combustion (90~150 yuan/ton) depends on air separation unit efficiency. (2) Scale effects—under high adoption (90% deployment), economies of scale could reduce unit costs by about 35%. (3) Energy price sensitivity—a 0.1 yuan/kWh electricity price increase raises operating costs by about 15 yuan/ton. (4) Technological advancement—emerging solvents (e.g., phase-change absorbers) may cut regeneration energy use by 20%~30%, though their commercial viability remains uncertain.

We estimate the operating costs of carbon capture under seven scenarios—Policy, Strengthened Policy, 2°C, 1.5°C, Literature-Based [12], Stated Policies (SPS), and Announced Pledges (APS)—for both low (30%) and high (90%) adoption rates in China's coal-fired power sector by 2050 (Tables 4 and 5, respectively). These costs align with the $CO_2$ reduction potential presented in Table 2 and account for technology-specific variations: pre-combustion capture (lower bound: 30~50 yuan/ton $CO_2$), post-combustion capture (80~150 yuan/ton), and oxy-fuel combustion (90~150 yuan/ton) [20]. While the calculation framework remains fixed, the cost ranges are designed to accommodate future adjustments reflecting technological advancements, such as improvements in solvent efficiency or energy recovery.

*Lower limit of pre combustion capture and emission reduction cost*
$$= 30\ (yuan/ton\ of\ CO2)\ *\ CO2\ reduction\ potential\ of\ coal$$
$$-\ fired\ power\ (billions\ of\ tons)\ *\ 3\%$$

(8)

*Upper limit of pre combustion capture and emission reduction cost*
$$= 50\ (yuan/ton\ of\ CO2)\ *\ CO2\ reduction\ potential\ of\ coal$$
$$-\ fired\ power\ (billions\ of\ tons)\ *\ 3\%$$

(9)

**Table 4. Operational costs of low-proportion carbon capture in China's coal power in 2050.**

| Scenarios | Policy scenarios | Strength-ened policy scenarios | 2°Csce-narios | 1.5°Csce-narios | Litera-ture [12] scenarios | SPS scenario | APS scenario |
|---|---|---|---|---|---|---|---|
| Lower operating cost limit of carbon capture and emission reduction for coal-fired power generation in 2050 (100 million yuan) - pre combustion capture (3%) | 8.84 | 6.36 | 1.09 | 0.27 | 2.50 | 7.57 | 3.66 |
| Upper operating cost limit of carbon capture and emission reduction for coal-fired power generation in 2050 (100 million yuan) - pre combustion capture (3%) | 14.73 | 10.61 | 1.82 | 0.45 | 4.17 | 12.62 | 6.11 |
| Lower operating cost limit of carbon capture and emission reduction for coal-fired power generation in 2050 (100 million yuan - post combustion capture (90%)) | 707.04 | 509.04 | 87.12 | 21.6 | 200.16 | 605.52 | 293.04 |
| Upper operating cost limit of carbon capture and emission reduction for coal-fired power generation in 2050 (100 million yuan - post combustion capture (90%)) | 1325.7 | 954.45 | 163.35 | 40.5 | 375.3 | 1135.35 | 549.45 |
| Lower operating costs limit for carbon capture and emission reduction in coal-fired power generation by 2050 (100 million yuan -7% for oxygen rich combustion capture) | 61.87 | 44.54 | 7.62 | 1.89 | 17.51 | 52.98 | 25.64 |
| Upper operating costs limit for carbon capture and emission reduction in coal-fired power generation by 2050 (100 million yuan -7% for oxygen rich combustion capture) | 103.11 | 74.24 | 12.71 | 3.15 | 29.19 | 88.31 | 42.74 |
| Total lower operating cost limit of coal-fired power carbon capture and emission reduction in 2050 (in billions of yuan) | 777.75 | 559.94 | 95.83 | 23.76 | 220.17 | 666.07 | 322.34 |
| Total upper operating cost limit of coal-fired power carbon capture and emission reduction in 2050 (in billions of yuan) | 1443.54 | 1039.3 | 177.88 | 44.1 | 408.66 | 1236.28 | 598.3 |

**Table 5. Operational costs of high-proportion carbon capture in China's coal power in 2050.**

| Scenarios | Policy scenarios | Strength-ened policy scenarios | 2°Csce-narios | 1.5°Csce-narios | Litera-ture [12] scenarios | SPS scenario | APS scenario |
|---|---|---|---|---|---|---|---|
| Lower operating cost limit of carbon capture and emission reduction for coal-fired power generation in 2050 (100 million yuan) - pre combustion capture (3%) | 26.52 | 19.09 | 3.27 | 0.81 | 7.51 | 22.71 | 10.99 |
| Upper operating cost limit of carbon capture and emission reduction for coal-fired power generation in 2050 (100 million yuan) - pre combustion capture (3%) | 44.19 | 31.82 | 5.45 | 1.35 | 12.51 | 37.85 | 18.32 |
| Lower operating cost limit of carbon capture and emission reduction for coal-fired power generation in 2050 (100 million yuan - post combustion capture (90%)) | 2121.12 | 1527.12 | 261.36 | 64.8 | 600.48 | 1816.56 | 879.12 |
| Upper operating cost limit of carbon capture and emission reduction for coal-fired power generation in 2050 (100 million yuan - post combustion capture (90%)) | 3977.1 | 2863.35 | 490.05 | 121.5 | 1125.9 | 3406.05 | 1648.35 |
| Lower operating costs limit for carbon capture and emission reduction in coal-fired power generation by 2050 (100 million yuan -7% for oxygen rich combustion capture) | 185.60 | 133.62 | 22.87 | 5.67 | 52.54 | 158.95 | 76.92 |
| Upper operating costs limit for carbon capture and emission reduction in coal-fired power generation by 2050 (100 million yuan -7% for oxygen rich combustion capture) | 309.33 | 222.71 | 38.12 | 9.45 | 87.57 | 264.92 | 128.21 |
| Total lower operating cost limit of coal-fired power carbon capture and emission reduction in 2050 (in billions of yuan) | 2333.25 | 1679.82 | 287.49 | 71.28 | 660.51 | 1998.21 | 967.02 |
| Total upper operating cost limit of coal-fired power carbon capture and emission reduction in 2050 (in billions of yuan) | 4330.62 | 3117.9 | 533.64 | 132.3 | 1225.98 | 3708.84 | 1794.9 |

$$\begin{aligned}
&\textit{The lower limit of post combustion capture and emission reduction cost is } 80 \\
&(\textit{yuan/ton of CO}2) \ast \textit{CO}2 \textit{ reduction potential of coal} - \textit{fired power (billions of tons)} \ast 90\%
\end{aligned} \tag{10}$$

$$\begin{aligned}
&\textit{Upper limit of post combustion capture and emission reduction cost} \\
&= 150 \ (\textit{yuan/ton of CO}2) \ast \textit{Potential for CO}2 \textit{ reduction from coal} \\
&- \textit{fired power (billions of tons)} \ast 90\%
\end{aligned} \tag{11}$$

$$\begin{aligned}
&\textit{Lower limit of cost for oxygen rich capture and emission reduction} \\
&= 90 \ (\textit{yuan/ton of CO}2) \ast \textit{Potential for CO}2 \textit{ reduction from coal} \\
&- \textit{fired power generation (billions of tons)} \ast 7\%
\end{aligned} \tag{12}$$

$$\begin{aligned}
&\textit{Upper limit of cost for oxygen rich capture and emission reduction} \\
&= 150 \ (\textit{yuan/ton of CO}2) \ast \textit{Potential for CO}2 \textit{ reduction from coal} \\
&- \textit{fired power (billions of tons)} \ast 7\%
\end{aligned} \tag{13}$$

$$\begin{aligned}
&\textit{The total operating cost of carbon capture and emission reduction in coal} - \textit{fired power plants} \\
&= \textit{pre combustion capture and emission reduction cost} \\
&+ \textit{post combustion capture and emission reduction cost} \\
&+ \textit{enriched oxygen capture and emission reduction cost}
\end{aligned} \tag{14}$$

### 4.3. Prediction of carbon capture investment scale in China's coal-fired power industry before 2050

China's coal-fired power fleet totaled 1.08 billion kilowatts by the end of 2020, comprising subcritical (356 MW, 32.9%), supercritical (285 MW, 26.4%), ultra-supercritical (250 MW, 23.1%), and substandard subcritical units (190 MW, 17.6%) [17]. With over half of operational units constructed during 2005–2015 and assuming 30-year lifespans, the sector will face peak retrofit demands between 2035 and 2045. Current projections indicate a phased technology transition: first-generation capture systems will dominate retrofits of existing plants before 2035, while second-generation systems will be deployed in new builds thereafter [21]. This aligns with China's power sector roadmap - coal capacity is expected to peak by 2025 under the 14th Five-Year Plan, with no new coal projects post-2030 and gradual retirement of aging units [21,22]. Carbon capture deployment is anticipated to scale from R&D-focused pre-2030 efforts to commercial implementation at 14 GW/year (280 GW cumulative by 2050), contingent on post-peak emission policies [22].

This study models the maturation pathway for coal-fired carbon capture technology in China, projecting large-scale deployment commencing post-2030 [11]. We evaluate seven distinct scenarios - Policy, Strengthened Policy, 2°C, 1.5°C, Literature-Based [12], Stated Policies (SPS), and Announced Pledges (APS) - through comprehensive cost analysis. The annual investment requirements (summing installation and operational costs) are presented for both conservative (30% adoption) and ambitious (90% adoption) pathways in Tables 6 and 7 respectively. These projections derive from: (1) 2050 coal power capacity estimates (Table 3), [2] technology-specific installation costs (Table 3), and (3) operational expenditures across capture methods (Tables 4 and 5). Our methodology enables dynamic adjustment of cost parameters while maintaining a consistent analytical framework for cross-scenario comparison.

**Table 6. Low adoption scenarios in 2050, investment in coal-fired power carbon capture.**

| Scenarios | Policy scenarios | Strengthened policy scenarios | 2°Cscenarios | 1.5°Cscenarios | Literature [12] scenarios | SPS scenario | APS scenario |
|---|---|---|---|---|---|---|---|
| Low adoption scenario in 2050, coal-fired power installed capacity, new carbon capture capacity (GW) | 11.595 | 8.745 | 1.845 | 0.48 | 4.5 | 15.66 | 7.59 |
| The installed capacity of coal-fired power in the 2050 low adoption scenario already has carbon capture capacity (GW) | 220.305 | 166.155 | 35.055 | 9.12 | 85.5 | 297.54 | 144.21 |
| Low adoption scenario in 2050, coal-fired power installed capacity, new carbon capture installation cost (100 million yuan) | 794.14~2382.42 | 598.95~1796.84 | 126.36~379.09 | 32.88~98.63 | 30.82~92.46 | 1072.55~3217.66 | 519.84~1559.52 |
| Operating cost of coal-fired carbon capture in 2050 (in billions of yuan) | 777.75~1443.54 | 559.94~1039.3 | 95.83~177.88 | 23.76~44.1 | 220.17~408.66 | 666.07~1236.28 | 322.34~598.3 |
| Investment cost of coal-fired power carbon capture under low adoption scenario in 2050 (in billions of yuan) | 1571.89~3825.96 | 1158.84~2836.14 | 222.19~556.97 | 56.64~142.73 | 250.99~501.12 | 1738.62~4453.94 | 842.18~2157.82 |

We quantify annual carbon capture investments from 2030 to 2050 across seven scenarios for both conservative (30%) and ambitious (90%) adoption pathways, with detailed results visualized in Figs 3–6. Table 8 presents the cumulative 2050 investments and operating costs, calculated as the sum of technology installation and operational expenditures for each scenario. These comprehensive cost assessments provide the foundation for future refinements incorporating financial parameters such as discount rates, which would enhance the economic precision of long-term deployment projections.

The large-scale deployment of CCUS technology faces significant financial risks, including potential cost overruns and implementation delays. A robust economic assessment framework incorporating Return on Investment (ROI), Net Present Value (NPV), and Internal Rate of Return (IRR) metrics is essential for evaluating project viability and long-term financial sustainability. ROI analysis helps determine whether projected returns justify the substantial upfront investments and operational expenses characteristic of CCUS systems, while NPV and IRR provide complementary perspectives on financial performance. Furthermore, policy mechanisms such as carbon pricing, targeted subsidies, and tax incentives can substantially improve the economic attractiveness of CCUS projects. This multidimensional analytical approach enables policymakers and investors to better balance technological risks against potential economic benefits, thereby facilitating more informed decisions regarding CCUS deployment at scale.

Comparative analysis of Figs 3–6 reveals distinct investment patterns across seven decarbonization scenarios: Policy, Strengthened Policy, 2°C, 1.5°C, Literature-Based [12], Stated Policies (SPS), and Announced Pledges (APS). The SPS scenario requires the highest cumulative investment in carbon capture deployment, while the 1.5°C pathway

Table 7. High adoption scenarios in 2050, investment in coal-fired power carbon capture.

| Scenarios | Policy scenarios | Strengthened policy scenarios | 2°Cscenarios | 1.5°Cscenarios | Literature [12] scenarios | SPS scenario | APS scenario |
|---|---|---|---|---|---|---|---|
| High adoption scenario in 2050, coal-fired power installed capacity, new carbon capture capacity (GW) | 34.785 | 26.235 | 5.535 | 1.44 | 13.5 | 46.98 | 22.77 |
| The installed capacity of coal-fired power in the 2050 high adoption scenario already has carbon capture capacity (GW) | 660.915 | 498.465 | 105.165 | 27.36 | 256.5 | 892.62 | 432.63 |
| High adoption scenario in 2050, coal-fired power installed capacity, new carbon capture installation cost (100 million yuan) | 2382.42~7147.27 | 1796.84~5390.51 | 379.09~1137.28 | 98.63~295.88 | 924.62~2773.85 | 3217.66~9652.98 | 1559.52~4678.55 |
| Operating cost of coal-fired carbon capture in 2050 (in billions of yuan) | 2333.25~4330.62 | 1679.82~3117.9 | 287.49~533.64 | 71.28~132.3 | 660.51~1225.98 | 1998.21~3708.84 | 967.02~1794.9 |
| Investment cost of coal-fired power carbon capture in high adoption scenarios by 2050 (in billions of yuan) | 4715.67~11477.89 | 3476.66~8508.41 | 666.58~1670.92 | 169.91~428.18 | 1585.13~3999.83 | 5215.87~13361.82 | 2526.54~6473.45 |

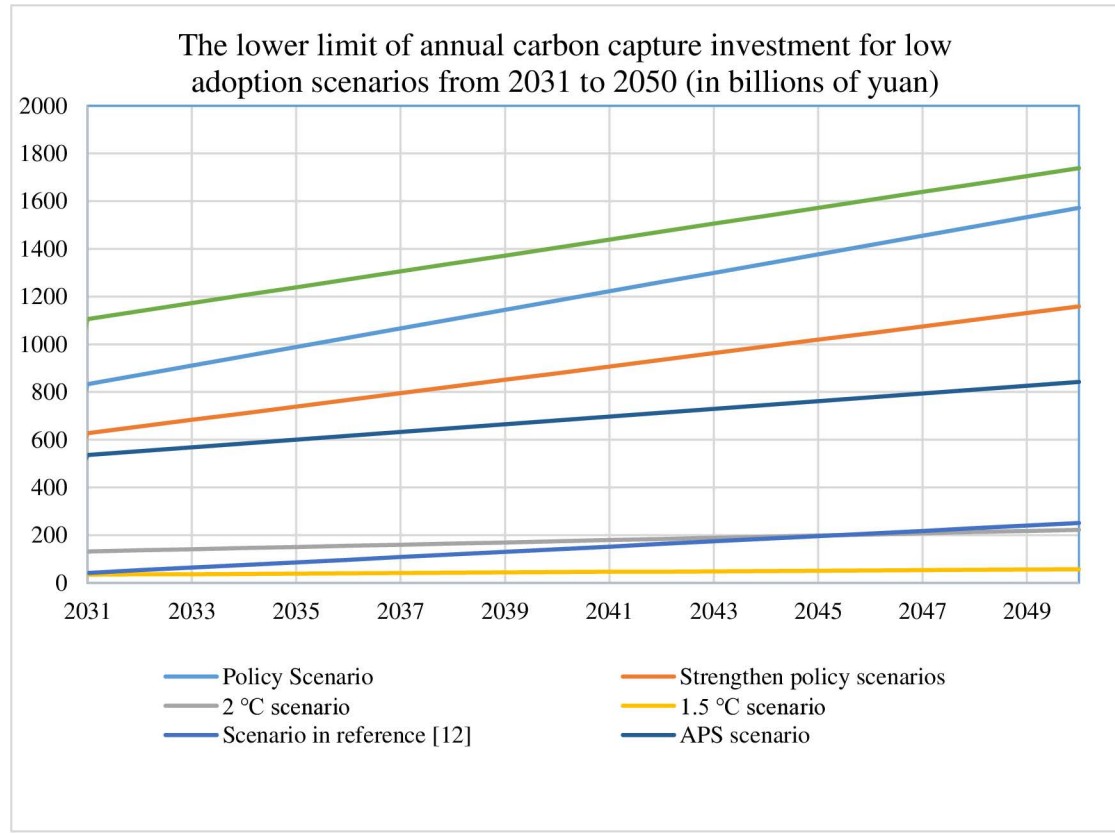

**Fig 3. Lower limit of annual carbon capture investment for low adoption scenarios from 2030 to 2050.**

demonstrates the lowest financial requirements. Our projections show approximately linear growth in investment needs under current modelling assumptions, though real-world deployment rates may accelerate due to technological break-throughs (e.g., cost reductions in capture systems) or strengthened policy support. This projected scaling of CCUS infra-structure - while methodologically consistent in our calculations - would necessitate concurrent enhancements in energy security measures and grid stabilization technologies to maintain system reliability during the transition period.

## 5. Discussions

### 5.1. Key findings of this study

This study evaluates China's coal power sector through seven decarbonization scenarios (Policy, Strengthened Policy, 2°C, 1.5°C, Literature-Based [12], SPS, and APS), projecting carbon capture potential, costs, and investment require-ments through 2050. Key findings reveal: (1) Substantial variation in emission reduction potential and costs, with 2050 investments ranging from 180~450 billion yuan under low adoption (30% capture) to 500 billion~1.35 trillion yuan under high adoption (90%), demonstrating policy intensity's direct impact on both climate and economic outcomes; (2) Deploy-ment pace critically influences feasibility - while 14 GW/year expansion could achieve 280 GW cumulative capacity by 2050, the high-adoption scenario's operational costs (80~150 yuan/ton $CO_2$) may challenge plant economics; (3) Marked scenario dependence, where ambitious climate targets (1.5°C/Strengthened Policy) reduce CCUS demand through coal phase-outs, whereas coal-reliant pathways (APS/SPS) require massive CCUS investment to meet emission goals. These results highlight the technology's contextual sensitivity across China's possible energy futures.

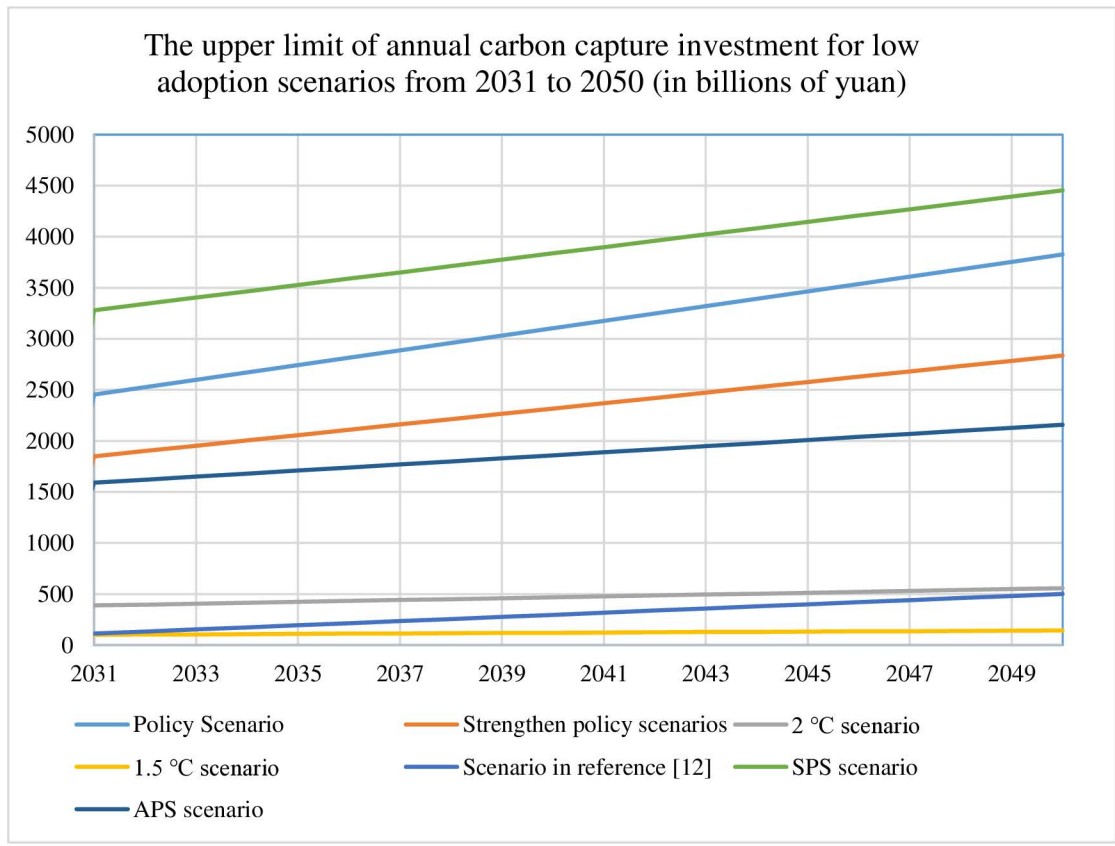

**Fig 4. Upper limit of annual carbon capture investment for low adoption scenarios from 2030 to 2050.**

## 5.2. Confidence level of predictive statistics

The substantial investment range projections (e.g., 91~450 billion yuan for the 30% adoption scenario) reflect significant parametric uncertainties in CCUS deployment economics. Three key factors dominate this variability: (1) installation costs could decline by 40%~50% by 2050 assuming a 10%~15% learning rate, (2) operational costs may reach their lower bound (30 yuan/ton $CO_2$) through energy efficiency improvements or advanced solvent technologies, and (3) policy implementation risks that could alter current cost trajectories. While these projections carry moderate confidence, they remain particularly sensitive to technological breakthroughs and regulatory developments in China's power sector.

To enhance forecast robustness, we recommend: (1) dynamic model updating incorporating observed technological learning rates (5%~10% annual cost reduction) and policy changes, (2) comprehensive sensitivity analyses for critical variables including carbon price scenarios (100~300 yuan/ton) and coal plant retirement schedules, and (3) probabilistic visualization of investment distributions through box plots or kernel density estimates. This multi-pronged approach would better characterize the risk-reward profile of large-scale CCUS deployment while informing adaptive policy design.

## 5.3. Comparison with international CCUS deployment cases

On Technological and Policy Gaps in CCUS Development, significant disparities exist between China and developed nations in CCUS technology maturity, economic viability, and policy frameworks. While countries like those in the EU and U.S. have established commercial-scale operations through integrated technology chains and market-based

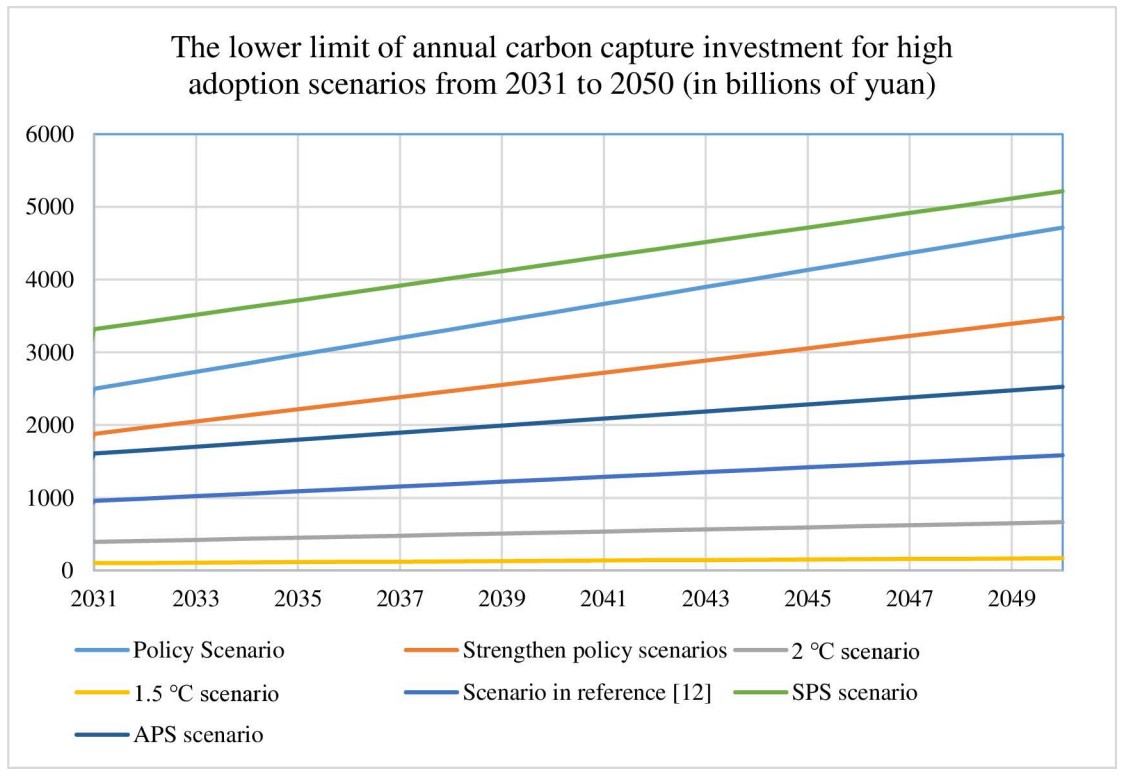

**Fig 5. Lower limit of annual carbon capture investment for high adoption scenarios from 2030 to 2050.**

mechanisms [23], China remains in the demonstration phase. For instance, Europe's Northern Lights project and the U.S. Decatur facility achieve million-ton-scale annual capture, whereas China's flagship Guoneng Taizhou project currently operates at 0.5 million tons/year capacity. These gaps stem from differences in both technological readiness and policy approaches - developed economies leverage carbon markets (e.g., EU ETS) and tax incentives (e.g., $85/ton under the U.S. Inflation Reduction Act), while China predominantly relies on direct fiscal subsidies without robust market drivers.

The European CCUS Roadmap: The European Green Deal positions CCUS as critical infrastructure for achieving 2050 carbon neutrality [24–26]. Projections indicate Europe's $CO_2$ capture capacity will scale from 2.3~4.3 million tons/year (2030) to 9.3~12 million tons/year (2050) under 1.5°C scenarios [25]. This growth reflects strategic policy commitments to accelerate project deployment and infrastructure development during 2020~2030, including cross-border $CO_2$ transport networks and storage hubs. The EU's multipronged approach combines regulatory mandates (e.g., revised ETS), innovation funding (Horizon Europe), and industrial partnerships to de-risk investments.

China's CCUS development faces three key challenges: (1) technological reliance on pilot-scale systems versus commercial-ready solutions, (2) limited market mechanisms to incentivize private investment, and (3) underdeveloped transport and storage infrastructure. The European experience demonstrates that combining carbon pricing ($85~100/ton), targeted R&D support, and coordinated infrastructure planning can overcome early-stage barriers. For China to bridge this gap, policy reforms should prioritize: establishing a national carbon market with CCUS inclusion, developing $CO_2$ pipeline networks in industrial clusters, and implementing production tax credits comparable to international benchmarks. These measures would accelerate China's transition from demonstration projects to commercially sustainable deployment at climate-relevant scales.

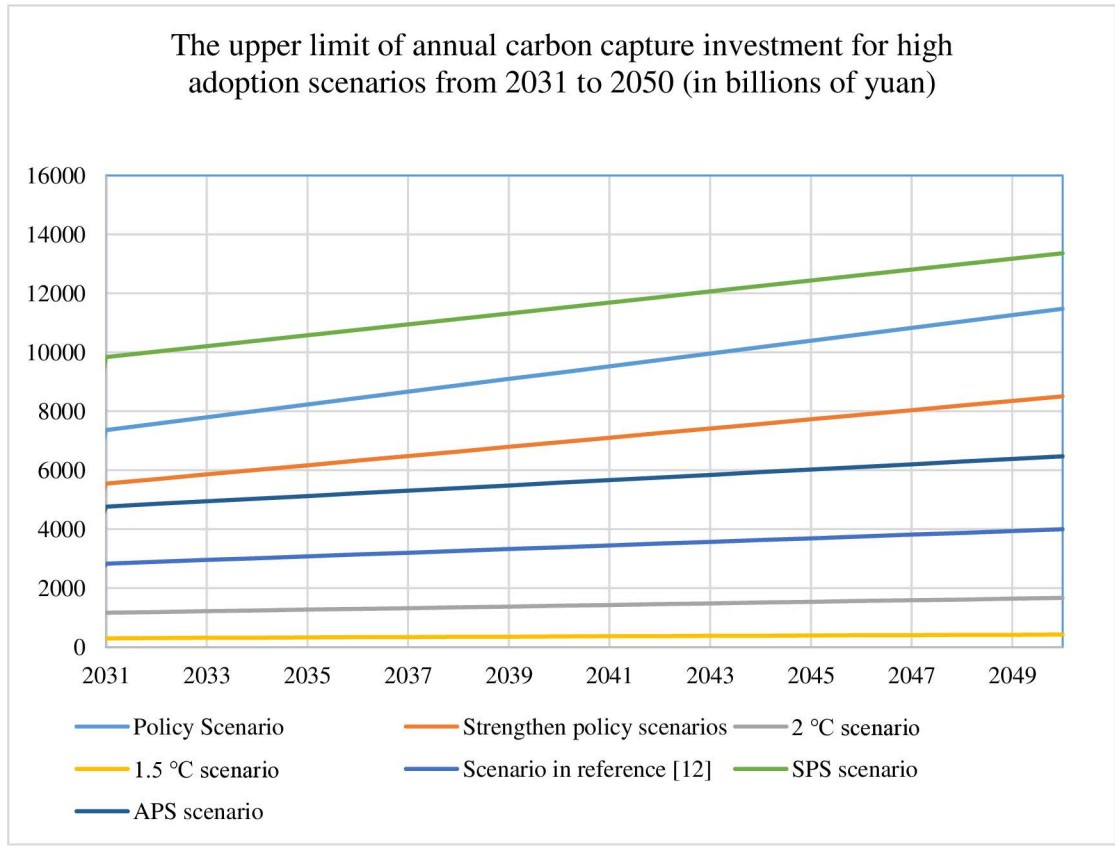

**Fig 6. Upper limit of carbon capture investment per year for high adoption scenarios from 2030 to 2050.**

## 5.4. Feasibility analysis of expected deployment scale

This study's assumption of 14 GW/year CCUS deployment in China's coal sector (280 GW by 2050) faces four key implementation challenges: (1) Technological readiness – while rapid post-2030 maturation is projected, actual progress depends on demonstration project outcomes, cost reduction rates (5%~10%/year needed), and policy stability, with international experience showing sustained carbon pricing (>50/ton) and subsidies are prerequisites for scale up; (2) Economic viability, retrofit costs(US$1000~3000/kw) and operating expenses (30~150 yuan/ton $CO_2$) may render coal power uncompetitive against renewables without robust carbon markets or fiscal support; (3) Sectoral transition – coal's evolving role toward peak-shaving (reducing capacity factors to 30%~40%) could undermine CCUS economics, while accelerated coal phase-outs under 1.5°C pathways may render 280GW targets obsolete; and (4) Regulatory gaps – absent comprehensive legislation governing $CO_2$ property rights, storage liability, and cross-ministerial coordination, project delays are likely. These interdependent constraints suggest the need for adaptive deployment targets tied to real-world policy and technology milestones.

## 5.5. Possible alternatives to CCUS

Complementary decarbonization pathways to CCUS include: (1) large-scale renewable energy deployment (wind, solar PV, hydropower) for direct fossil fuel substitution, avoiding downstream capture requirements; (2) system-wide energy efficiency improvements in industrial processes and buildings to reduce baseline emissions; (3) negative-emission bioenergy systems (BECCS) contingent on sustainable biomass availability; and (4) enabling technologies like green hydrogen

Table 8. Accumulated investment in carbon capture of coal-fired power in China under 7 scenarios by 2050.

| Scenarios | Policy scenarios | Strengthened policy scenarios | 2°C scenarios | 1.5°C scenarios | Literature [10] scenarios | SPS scenario | APS scenario |
|---|---|---|---|---|---|---|---|
| Accumulated operating cost of low adoption scenario (100 million yuan) | 8166.38~15157.17 | 5879.37~10912.65 | 1006.22~1867.74 | 249.48~463.05 | 2311.79~4290.93 | 6993.74~12980.94 | 3384.57~6282.15 |
| Accumulated operating cost of high adoption scenarios (100 million yuan) | 24499.13~45471.51 | 17638.11~32737.95 | 3018.65~5603.22 | 748.44~1389.15 | 6935.36~12872.79 | 20981.21~38942.82 | 10153.71~18846.45 |
| Accumulated investment cost for low adoption scenarios (100 million yuan) | 24049.38~62805.17 | 17858.37~46849.65 | 3533.22~9449.74 | 907.48~2400.05 | 8475.79~22782.93 | 28444.74~77333.94 | 13781.57~37472.15 |
| Accumulated investment cost for high adoption scenarios (in billions of yuan) | 72147.13~188416.51 | 53575.11~140547.95 | 10600.65~28349.22 | 2721.44~7307.15 | 25427.36~68349.79 | 85334.21~232002.82 | 41343.71~112417.45 |

production, advanced energy storage, and smart grid integration to maximize clean energy utilization. When combined with natural climate solutions (e.g., enhanced forest/soil carbon sequestration) and circular economy practices (e.g., carbon-cycling materials), these strategies can collectively achieve carbon neutrality with potentially lower costs and greater sustainability than CCUS-dependent pathways. Critical implementation challenges include the intermittent nature of renewables (requiring 80%~120% grid flexibility increases), BECCS land-use constraints (3%~5% of agricultural land per Gt $CO_2$ stored), and hydrogen's current cost premiums ((2–3)× conventional fuels).

### 5.6. Sensitivity analysis for investment forecasting

Sensitivity analysis reveals four critical factors influencing coal-fired CCUS investment forecasts in China: (1) Technology cost volatility – with installation costs ($1,000~3,000/kW) and operating costs (30~150 yuan/ton $CO_2$) as key determinants, where a 30% cost reduction could decrease cumulative investments by 20%~35%, while cost escalations would impair project viability; (2) Policy scenario divergence – high deployment (90%) requires 23.2 trillion yuan (5×low deployment's 30% scenario), demonstrating policy's pivotal role in capital allocation; (3) Market mechanisms – carbon prices exceeding 200 yuan/ton or electricity subsidies could boost project IRR by 3~5 percentage points, substantially improving bankability; and (4) Coal power trajectory uncertainty – accelerated retirements in ambitious climate scenarios (e.g., 1.5°C) may reduce CCUS demand by 50%, whereas coal-reliant pathways (e.g., SPS) necessitate greater investment. To strengthen predictive robustness, future modelling should integrate dynamic carbon pricing, regional heterogeneities, and technology learning curves (typically 10%~15% cost reduction per doubling of capacity).

### 5.7. The role of CCUS technology in achieving China's "dual carbon" goals

Strategic Importance of CCUS for China's Decarbonization: CCUS technology represents a critical pathway for China to achieve its carbon peaking (2030) and neutrality (2060) targets, particularly for hard-to-abate sectors like coal-fired power. Projections indicate it could deliver 15%~32% of cumulative emission reductions, with high-deployment scenarios potentially mitigating 1.2~2.9 billion tons of $CO_2$ annually by 2050. Beyond emissions mitigation, CCUS supports three key transition objectives: (1) ensuring energy security through reliable low-carbon baseload power during renewable integration, (2) catalyzing a 20+trillion yuan industrial chain with associated green employment, and (3) establishing infrastructure for future negative-emission technologies like BECCS. Realizing this potential requires concurrent policy advancements (carbon prices >200 yuan/ton) and technological innovations (50% cost reductions).

Implementation Requirements and Economic Synergies: The economic viability of CCUS hinges on cross-sectoral value creation, including enhanced oil recovery (increasing yields by 15%~20%) and chemical feedstock production. However, large-scale deployment faces dual challenges: (1) infrastructure demands, requiring 10,000+km of $CO_2$ pipelines and 50~100 major storage sites by 2050, and (2) market design needs, particularly flexible electricity pricing mechanisms to accommodate capture systems' energy penalties (20%~30% plant output). International benchmarks demonstrate that successful CCUS industrialization depends on integrated policy packages combining carbon pricing, capital subsidies (covering 30%~50% of retrofit costs), and liability frameworks for long-term storage.

Transferable Lessons for Coal-Dependent Economies: While this study's methodology provides a reference framework, implementation strategies must adapt to local contexts: (1) Policy instruments should align with governance systems - carbon markets suit market economies (EU), whereas administrative measures may dominate planned economies; (2) Geological potential dictates infrastructure planning, with Australia's sedimentary basins offering 200+years of storage capacity versus limited options in Japan; (3) Energy transition speeds determine CCUS relevance - Germany's rapid renewables expansion (50% by 2030) reduces coal dependence faster than India's; (4) Economic thresholds vary, where less than $100/ton carbon prices or less than 4,000 operating hours jeopardize project bankability. China's experience underscores that CCUS effectiveness scales with policy coordination, particularly in synchronizing carbon pricing with grid modernization investments.

## 5.8. Core challenges and policy considerations for implementing large-scale CCUS in China

The technical challenges of implementing CCUS in China's coal-fired power sector are multifaceted. Firstly, the cost bottleneck remains a critical issue, as post-combustion capture technology currently reduces power plant efficiency by 15%~25% and incurs high unit costs of 300~600 yuan/ton $CO_2$. Addressing this requires breakthroughs in key technologies such as advanced adsorbents to achieve cost reductions. Secondly, scaling presents a significant challenge, with existing demonstration projects capturing less than one million tons annually, while billion-ton deployments will be necessary by 2050. This gap underscores the urgent need to resolve engineering stability issues during large-scale implementation.

Key policy considerations must focus on creating an enabling environment for CCUS deployment. Establishing comprehensive economic incentives through a "carbon pricing + subsidies + green finance" policy framework is essential, with carbon prices needing to exceed 200 yuan/ton to ensure viability. Infrastructure development should be prioritized, particularly the construction of a North China-Northwest $CO_2$ transportation pipeline network to reduce marginal costs. Additionally, regulatory frameworks must be strengthened to clarify long-term storage responsibilities and resolve property rights disputes, providing the legal certainty required for large-scale investments. These measures collectively address both the technical and systemic barriers to CCUS implementation in China's energy transition.

## 5.9. The environmental impact and economic benefits of deploying CCUS in coal-fired power plants exhibit phased characteristics

The environmental impacts of CCUS deployment present both benefits and challenges. A key benefit is the potential for near-zero emissions from coal-fired power plants, with capture rates reaching 90% and reducing annual $CO_2$ emissions by approximately 2 million tons per unit. Additionally, CCUS systems can simultaneously decrease sulfur and nitrogen oxide emissions by 30%~50%, contributing to improved air quality. The sequestration of $CO_2$ also plays a crucial role in mitigating long-term climate risks and supporting carbon neutrality objectives. However, these benefits come with potential costs: CCUS implementation increases system energy consumption by 20%~30%, and reliance on coal plants for energy replenishment may temporarily raise grid carbon emissions by 5%~8%. Furthermore, the degradation of chemical absorbents could generate harmful by products like nitrosamines, necessitating stringent regulatory oversight.

The economic and environmental trade-offs of CCUS evolve over time. In the short term (pre-2030), high emission reduction costs (300~600 yuan/ton) result in negative net environmental benefits, requiring substantial policy subsidies (covering about 60% of costs). However, long-term projections (post-2040) indicate a shift toward synergy: technological advancements are expected to reduce costs to 150 yuan/ton, while grid decarbonization will lower lifecycle emissions by over 85%. This transition could align with the development of a 20 trillion yuan industrial sector, creating a sustainable balance between economic growth and emission reductions.

To reconcile early-stage challenges, a three-pronged strategy—combining green electricity capture, pipeline infrastructure, and carbon pricing—is essential. Analysis shows that under high deployment (90%), cumulative emission reductions (2.9 billion tons by 2050) can offset transformation costs. However, achieving this hinges on consistent policy support, particularly carbon pricing exceeding 200 yuan/ton, to ensure economic viability and environmental effectiveness.

## 5.10. Limitations of the study and suggestions

Technology Maturity Assumptions: The study's projection of rapid CCUS adoption post-2030 carries inherent uncertainties, as actual deployment rates depend on technological breakthroughs, policy incentives, and cost reduction trajectories. To enhance reliability, future assessments should incorporate dynamic modelling of promotion speeds using technology learning curves, while accounting for variations in development potential across different capture methods (e.g., post-combustion vs. oxy-fuel systems).

Cost Estimation Methodology: Current cost projections for 2050 rely on linear extrapolations and fixed proportions, overlooking critical factors such as non-linear technological advancements and regional disparities (e.g., between coastal and inland plants). A more robust approach would integrate sensitivity analyses that consider both technological learning rates (typically 10%~15% per capacity doubling) and region-specific variables like fuel prices and infrastructure costs.

Operational Realities: The analysis omits key operational constraints, including the 10%~30% efficiency penalty imposed by capture systems and associated grid flexibility requirements. Subsequent research should evaluate these factors in conjunction with evolving power plant utilization rates and potential synergies with energy storage technologies to provide a more accurate assessment of achievable emission reductions.

Data Constraints: The use of Shanghai-specific emission factors may underestimate outputs from high-emission units elsewhere in China. Refining this aspect requires adopting regionally differentiated emission factors that correlate with unit age and technology profiles, thereby improving the representativeness of national-scale projections.

## 6. Conclusion

This study evaluates China's coal-fired power sector through seven decarbonization scenarios, analyzing capacity trends, generation patterns, and CCUS deployment potential through 2050. The results demonstrate that high-proportion carbon capture adoption (90%) could effectively facilitate the industry's low-carbon transition, with cumulative investments ranging from 91~232 billion yuan before 2050. At scale, CCUS deployment may catalyze an emerging 20 + trillion yuan industry, simultaneously enabling emission reductions and green economic growth. However, significant policy barriers persist, including inadequate carbon pricing (current 70~80 vs. required 200 yuan/ton), unstable subsidy frameworks, and regulatory ambiguities regarding $CO_2$ pipeline infrastructure and storage site governance. These challenges, compounded by renewable energy expansion reshaping coal's role, risk delaying critical investment decisions without targeted interventions.

To realize CCUS's potential, China must implement a comprehensive policy framework encompassing: (1) clear technology roadmaps aligned with carbon neutrality targets, (2) enhanced R&D investment and standardized technology protocols, (3) stable financing mechanisms including long-term subsidies and carbon market integration, and (4) international collaboration on technical standards. Legislative actions should address infrastructure coordination and storage liabilities, while pilot programs could de-risk regional deployments. Such coordinated measures would optimize the dual benefits of emission mitigation and industrial development during the energy transition.

Our analysis reveals substantial emission reduction potential through CCUS deployment in China's coal power sector, with high-adoption scenarios (90% capture rate) capable of delivering billion-ton $CO_2$ reductions by 2050 - positioning CCUS as a critical carbon neutrality solution. The investment spectrum varies dramatically by adoption level: 91~450 billion yuan for conservative (30%) implementation versus 0.5~23.2 trillion yuan for ambitious (90%) scenarios, potentially catalyzing a 20 trillion-yuan green industry. However, cost sensitivities present significant barriers, particularly operational expenses (30~150 yuan/ton $CO_2$) and retrofitting costs ($1,000~3,000/kW), underscoring the need for targeted policy interventions to ensure economic viability.

Four priority actions emerge. First, develop phased CCUS roadmaps (2030–2050) integrating carbon markets and electricity pricing reforms, with dynamic technical-economic assessments guiding regional deployment. Second, establish national R&D initiatives targeting less than 50 yuan/ton capture costs through advanced post-combustion technologies while leading global standard development. Third, create innovative financing mechanisms including green bonds and carbon credits to mitigate early-stage risks. Fourth, implement demonstration clusters in critical regions (e.g., North/Northwest China) leveraging international expertise and participating in CDR initiatives to accelerate technology transfer and large-scale validation.

Urgent Collaborative Action Required. The current decade represents a critical juncture for establishing CCUS as a foundational - rather than supplementary - carbon neutrality solution. This demands coordinated action across three

fronts: (1) policymakers must accelerate institutional innovations to create stable regulatory frameworks, (2) researchers should intensify cross-disciplinary efforts to address key technical challenges (particularly the 30~150 yuan/ton operating cost barrier), and (3) industry must advance technology readiness through strategic pre-deployment. Our analysis indicates that such tripartite collaboration could mobilize the projected 23.2 trillion yuan investment into tangible climate and economic benefits, transforming CCUS into both an emission reduction mechanism and growth driver.

Validation and Refinement of Research Findings include: While confirming core hypotheses regarding CCUS scalability (evidenced by 2050 investment projections) and emission reduction efficacy, the study revealed unexpected complexities. The wider-than-anticipated investment range (91 billion~23.2 trillion yuan) underscores the outsized influence of policy scenarios, while the emergent 20 trillion-yuan industry potential exceeded initial expectations. However, certain assumptions require reevaluation, particularly the uniform 14GW/year deployment rate (potentially incompatible with 1.5°C pathways) and regional technological disparities. Future work should incorporate dynamic hypothesis testing and geographically differentiated analysis to enhance predictive accuracy as policies evolve.

## Supporting information

**S1 Data. Data calculation for the paper.**
(XLSX)

## Author contributions

**Conceptualization:** Jianfang Zong, Haitao Wang.

**Data curation:** Jianfang Zong, Xinyu Cao.

**Formal analysis:** Jianfang Zong, Qing Ding.

**Funding acquisition:** Jianfang Zong.

**Investigation:** Jianfang Zong, Jie Wei.

**Methodology:** Jianfang Zong.

**Project administration:** Jianfang Zong.

**Resources:** Jianfang Zong.

**Supervision:** Jianfang Zong.

**Writing – original draft:** Jianfang Zong.

**Writing – review & editing:** Jianfang Zong.

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
