## [Decision Letter · Decision Letter 0]

18 Mar 2025

PONE-D-25-00633Analysis of the deployment scale and investment prediction of China's coal power carbon capture technology under typical scenarios before 2050PLOS ONE

Dear Dr. Zong,

Thank you for submitting your manuscript to PLOS ONE. After careful consideration, we feel that it has merit but does not fully meet PLOS ONE’s publication criteria as it currently stands. Therefore, we invite you to submit a revised version of the manuscript that addresses the points raised during the review process.

We look forward to receiving your revised manuscript.

Kind regards,

Soheil Mohtaram

Academic Editor

PLOS ONE

[This work was financially supported by Research and Application of Carbon Accounting Methods for Financial Institutions under the Fund Project of the President of China Institute of Standardization, No. 542022Y-9372.]

 [The author(s) received no specific funding for this work.]

Reviewers' comments:

Reviewer's Responses to Questions

**Comments to the Author**

1. Is the manuscript technically sound, and do the data support the conclusions?

Reviewer #1: Yes

Reviewer #2: Yes

Reviewer #3: Partly

Reviewer #4: Yes

2. Has the statistical analysis been performed appropriately and rigorously? 

Reviewer #1: N/A

Reviewer #2: Yes

Reviewer #3: Yes

Reviewer #4: No

3. Have the authors made all data underlying the findings in their manuscript fully available?

Reviewer #1: Yes

Reviewer #2: Yes

Reviewer #3: Yes

Reviewer #4: Yes

4. Is the manuscript presented in an intelligible fashion and written in standard English?

Reviewer #1: Yes

Reviewer #2: Yes

Reviewer #3: Yes

Reviewer #4: Yes

5. Review Comments to the Author

Reviewer #1: This manuscript investigates a crucial area of the analysis of the deployment scale and investment prediction of china's coal power carbon capture technology under typical scenarios before 2050 , which has become increasingly crucial in a changing climate. The manuscript is worthy of publication in this journal, but the major revision listed below needs to be addressed before publication. Nonetheless, my concrete comments are given below manuscript section-wise, which are needed to solve and improve the manuscript’s quality.

Overall comments:

a. The presentation of the manuscript requires improvement, particularly regarding sentence structure and grammatical errors. Revising sentences for consistency, cohesion, and standardization is essential, as it will enhance the overall quality of the manuscript for readers.

b. Authors should read the whole manuscript several times after correcting the comments below section-wise with full concentration. Then, the whole manuscript needs to be refined/rewritten/reorganized sentences to maintain sentence consistency and subsequently easy understanding for scientific readership from the preceding sentence to the running sentence.

c. References should be checked for improvement with the currently published article.

Abstract

1. Authors should write about the innovation or new findings from this research. The authors should identify and mention the significant findings of the results here, rather than mentioning all the findings.

2. Where are the methods to predict the carbon emissions of coal-fired power under the seven scenarios? Where are the results delineated in this study? Authors should address these issues.

3. Where is the take-home message for policymakers? Authors should write at least one sentence about this.

Introduction

4. I have read the introduction several times to understand the research rationale, aims, and objectives. What authors need to do is rewrite/reorganize this section. The introduction should clearly state the reason for the study, the hypothesis, and the essential background with recent references. In this regard, the below-mentioned article needs to be looked at for how to present a manuscript’s introduction and cite scientific evidence. https://doi.org/10.1038/s41467-024-55332-5,
https://doi.org/10.1016/j.isci.2022.105664.

5. Does the transition between various background elements occur seamlessly and in a logically coherent manner? Is the introduction effective in establishing a clear and logical setting for the research problem? Are the reasons for examining carbon capture in China's coal power sector clearly defined?

6. Is the introduction sufficiently referencing pertinent literature on CCUS (Carbon Capture, Utilization, and Storage)? Are the discussions surrounding China's climate commitments and energy policies current?

7. Is there a well-defined research gap identified in the introduction that this study intends to address? Is the introduction effectively articulating the study's objectives and hypotheses?

8. CCUS should be elaborated first for each section, and then the short term of CCUS should be used.

Prediction and analysis of the development capacity of carbon capture projects

9. Is the reasoning behind the selection of the seven scenarios sufficiently articulated? Are the assumptions regarding cost prediction, specifically for installation and operational costs, clearly defined?

10. Is there an explanation provided in the study regarding the data sources utilized for the analysis of investment and carbon reduction potential? Is there adequate detail in the calculations and modeling techniques provided for replication purposes?

11. Does the study address any limitations or uncertainties associated with the predictive modeling approach? Are references supplied for all essential assumptions (e.g., CO₂ emission factors, cost estimations)? Would incorporating further validation or conducting a sensitivity analysis enhance the robustness of the methodological approach?

Results

Please recheck the results and calculation sheets to delineate the exact and accurate amount to be sure again.

12. For the deployment Scale of Carbon Capture, are the deployment trends accurately represented through the use of figures and tables? Are all essential numerical outcomes (e.g., gigawatts installed, CO₂ captured) adequately contextualized? Are the distinctions between scenarios clearly articulated and substantiated? Is there any discussion regarding technological feasibility in this section? Are the policy implications of the projected deployment trends discussed? Does the conversation address the impact of deployment scale on energy security or grid stability?

13. For Carbon emissions and Capture potential, Are the calculations providing strong support for the emission reduction estimates? Does the study take into account aspects like efficiency enhancements in coal-fired plants? Is the explanation of Table 1 and other data visualizations clear and comprehensive in the text?

14. Are scenarios with varying capture ratios (low versus high proportion) adequately compared? Does the study address the alignment of these emission reduction figures with China's carbon neutrality objectives? Are potential uncertainties in emissions estimates recognized?

15. Are there any conversations regarding the possibility of carbon leakage or constraints in capture efficiency? Can external validation enhance the reliability of the projections?

Prediction of carbon capture investment scale

16. Is there a clear distinction between installation and operational costs?

17. Does the study take into account learning curves or cost reductions as time progresses? Is there an analysis regarding the economic viability of various carbon capture scenarios?

18. Are Tables 3–5 displaying the data in a clear and accessible way?

19. Are the impacts of policy incentives, subsidies, or carbon pricing mechanisms examined in relation to investment requirements? Does the analysis consider potential cost overruns or delays in the deployment of CCUS? Would it be possible to incorporate additional economic indicators, such as return on investment (ROI)?

Discussion

20. Does the discussion successful in interpreting the key findings of the study?

21. Are there any comparisons drawn with other international cases of CCUS deployment?

22. Does the section provide a critical evaluation of the feasibility of reaching the anticipated deployment scales? Is there a discussion on the policy and regulatory barriers to the adoption of CCUS?

23. Does the discussion explore possible alternatives to CCUS, such as the expansion of renewable energy sources?

24. Where are the study limitations and recommendations missing? It should be addressed briefly.

Conclusion

25. Does the manuscript's conclusion adequately summarize its key findings and offer a stronger emphasis on actionable recommendations? Does the study end with a compelling call to action for researchers and policymakers?

26. Is the study hypothesis validated by the study findings? Authors should write about these issues.

Reviewer #2: The discussion should explore policy implications in greater detail, particularly in terms of policy interventions needed to achieve large-scale CCUS deployment.

Figures and tables are well-organized, but Figure captions should be more descriptive to improve standalone readability.

The conclusion could benefit from a stronger emphasis on practical recommendations for policymakers and industry stakeholders.

The statistical approach is sound overall, but:

The manuscript lacks a sensitivity analysis for investment predictions. Given the uncertainties in cost and policy changes, a sensitivity analysis would strengthen the conclusions.

More details should be provided on the statistical confidence of the projections.

Reviewer #3: his paper analyzes the deployment scale, carbon reduction potential, and investment prediction of carbon capture technology in China's coal power sector before 2050. Using seven typical scenarios for China's green and low-carbon transformation of coal-fired power, the study assesses the long-term evolution of coal power systems, including total installed capacity, power generation, and carbon emissions. Here are some questions based on the paper:

1. How does the paper define the seven typical scenarios used to model China's green and low-carbon transformation of coal-fired power?

2. What role does carbon capture technology (CCUS) play in achieving China's carbon neutrality and carbon peak targets by 2050?

3. Can the paper's findings be applied to other countries with high reliance on coal-fired power, or are they specific to China's energy policies and conditions?

4. What are the key differences between the low-proportion and high-proportion carbon capture installation scenarios?

5. How does the paper predict the investment required for carbon capture technology in the coal power sector by 2050, and what are the estimated costs?

6. How might the deployment of carbon capture technology in coal-fired power plants impact China's energy market and economy?

7. What are the technological challenges and policy considerations highlighted by the authors for implementing large-scale carbon capture in China?

8. What are the environmental implications of carbon capture technology deployment in coal-fired power plants, and how do these compare with the economic benefits outlined in the paper?

9. Literature review/ introduction section needs to be improved significantly. All the references are fairly old. It is suggested to add some recent reference related to this work, from 2023-2024.

Reviewer #4: The manuscript presents a comprehensive assessment of carbon capture technology deployment in China’s coal-fired power sector. The study evaluates various scenarios, predicting carbon emissions, capture potential, and investment needs before 2050. The topic is relevant given global carbon neutrality goals, and the manuscript is structured with a clear methodology, detailed scenario analysis, and quantitative predictions. However, some areas require clarification and improvement to enhance the study’s robustness.

Text: "The paper also predicts and analyzes the installation cost, operating cost, and investment scale of coal-fired carbon capture facilities before 2050 under seven scenarios." (Page 1)

→ Consider explicitly stating the key assumptions behind cost calculations to improve transparency.

Text: "CCUS technology is a key technology for achieving low-carbon utilization of fossil fuels and supporting low-carbon emissions in industrial processes such as coal-fired power." (Page 1)

→ Clarify whether the study focuses on specific CCUS technologies (e.g., post-combustion, pre-combustion) or a general assessment.

Text: "The International Energy Agency (IEA) also estimates that the total amount of CO2 captured using carbon capture technology globally will increase from approximately 40 million tons in 2020 to approximately 7.6 billion tons by 2050." (Page 2)

→ Cite the specific IEA report and year for accurate reference.

Text: "According to different engineering techniques, it can be divided into CO2 geological utilization, CO2 chemical utilization, and CO2 biological utilization." (Page 3)

→ Provide specific examples or case studies for each category to improve clarity.

Text: "This article focuses on studying the carbon capture capacity, carbon reduction potential, and investment demand prediction of coal-fired power generation." (Page 3)

→ Explain how these predictions compare with international benchmarks.

Text: "Table 1. Carbon capture emissions from coal-fired power in China under seven scenarios by 2050" (Page 5)

→ Some numerical values lack explanations. Consider adding a brief discussion on the variations between scenarios.

Text: "Based on this data, calculate the installation cost of carbon capture for coal-fired power in China under seven scenarios in 2050." (Page 7)

→ Clarify the methodology used for cost estimation (e.g., exchange rate assumptions, inflation considerations).

Text: "Table 4. Operating costs for carbon capture and emission reduction in China's coal-fired power plants under 7 scenarios in 2050" (Page 9)

→ The operating cost range is wide. It would be useful to explain the key cost drivers leading to this variation.

Text: "Reference [13] points out that China will promote CCUS technology on a large scale from 2050 to 2060, until it covers all coal-fired power units." (Page 10)

→ Justify why large-scale adoption is expected to occur after 2050 and not earlier.

Text: "The widespread deployment of coal-fired carbon capture technology may create an emerging industry with a scale of over 20 trillion yuan, achieving a win-win situation for the low-carbon transformation of coal-fired power and green economic growth." (Page 13)

→ Consider discussing potential policy challenges that could hinder this large-scale adoption.

6. PLOS authors have the option to publish the peer review history of their article (what does this mean? ). If published, this will include your full peer review and any attached files.

**Do you want your identity to be public for this peer review?** For information about this choice, including consent withdrawal, please see our Privacy Policy .

Reviewer #1: **Yes: ** Md Mashiur Rahman

Reviewer #2: No

Reviewer #3: No

Reviewer #4: **Yes: ** Waqas A. Cheema

---

## [Author Response · Author response to Decision Letter 1]

28 Mar 2025

Thank you very much to the reviewers, your comments are very valuable for improving my manuscript, and I have made changes to all the comments that you have raised to me.

---

## [Decision Letter · Decision Letter 1]

23 Apr 2025

Analysis of the deployment scale and investment prediction of China's coal power carbon capture technology under typical scenarios before 2050

PONE-D-25-00633R1

Dear Dr. Jianfang Zong,

We’re pleased to inform you that your manuscript has been judged scientifically suitable for publication and will be formally accepted for publication once it meets all outstanding technical requirements.

Kind regards,

Soheil Mohtaram

Academic Editor

PLOS ONE

Reviewers' comments:

Reviewer's Responses to Questions

**Comments to the Author**

If the authors have adequately addressed your comments raised in a previous round of review and you feel that this manuscript is now acceptable for publication, you may indicate that here to bypass the “Comments to the Author” section, enter your conflict of interest statement in the “Confidential to Editor” section, and submit your "Accept" recommendation.

Reviewer #1: All comments have been addressed

Reviewer #2: All comments have been addressed

Reviewer #3: All comments have been addressed

 Review Comments to the Author

Reviewer #1: I am pleased that the authors responded to all the comments. The response to the comments provided by the authors is sufficient and could be considered for publishing this manuscript. Furthermore, the responses to the comments in the introduction have resulted in a more scientifically readable manuscript; corrections in the methodology and result discussion sections and the provided conclusion section are now consistently organized and rewritten sufficiently

Reviewer #2: Thank you very much for addressing all the comments. The major issue i could not understand that what is the main purpose behind this study , what is the novelty and main objectives and what is its impact on the public and audience.

Graphical abstract is missing, proper methodology flow chart is missing , check grammatical errors and reformat according to the journal formatting guidelines

Reviewer #3: Author addressed all the comments and incorporated the required changes. I have no more comments about this.

---

## [Editor Report · Acceptance letter]

PONE-D-25-00633R1

PLOS ONE

Dear Dr. Zong,

I'm pleased to inform you that your manuscript has been deemed suitable for publication in PLOS ONE. Congratulations! Your manuscript is now being handed over to our production team.

Kind regards,

on behalf of

Dr. Soheil Mohtaram

Academic Editor

PLOS ONE